# Relative Validity and Reproducibility of a Web-Based Semi-Quantitative Food Frequency Questionnaire in the Danish Diet, Cancer, and Health—Next Generations MAX Study

**DOI:** 10.3390/nu15102389

**Published:** 2023-05-19

**Authors:** Agnetha Linn Rostgaard-Hansen, Susanne Rosthøj, Carl Brunius, Sjurdur Frodi Olsen, Anne Ahrendt Bjerregaard, Janet Elisabeth Cade, Anne Tjønneland, Rikard Landberg, Jytte Halkjær

**Affiliations:** 1Department Life Sciences, Division of Food and Nutrition Science, Chalmers University of Technology, 41296 Gothenburg, Sweden; carl.brunius@chalmers.se (C.B.); rikard.landberg@chalmers.se (R.L.); 2Danish Cancer Society Research Center, 2100 Copenhagen, Denmark; suro@cancer.dk (S.R.); annet@cancer.dk (A.T.); 3Centre for Fetal Programming, Department of Epidemiology Research, Statens Serum Institut, 2300 Copenhagen, Denmark; sfo@ssi.dk (S.F.O.); anne.ahrendt.bjerregaard@regionh.dk (A.A.B.); 4Department of Public Health, University of Copenhagen, 1353 Copenhagen, Denmark; 5Center for Clinical Research and Prevention, Frederiksberg and Bispebjerg Hospital, 2000 Frederiksberg, Denmark; 6Nutritional Epidemiology, School of Food Science and Nutrition, University of Leeds, Leeds LS2 9JT, UK; j.e.cade@leeds.ac.uk

**Keywords:** food frequency questionnaire, 24-h dietary recall, web-based, relative validity, reproducibility, epidemiology, nutrition, diet, cancer

## Abstract

The food frequency questionnaire (FFQ) is designed to capture an individual’s habitual dietary intake and is the most applied method in nutritional epidemiology. Our aim was to assess the relative validity and reproducibility of the FFQ used in the Diet, Cancer, and Health—Next Generations cohort (DCH-NG). We included 415 Danish women and men aged 18–67 years. Spearman’s correlations coefficients, Bland–Altman limits of agreement and cross-classification between dietary intakes estimated from the FFQ administered at baseline (FFQ_baseline_), and the mean of three 24-h dietary recalls (24-HDRs) and the FFQ administered after 12 months (FFQ_12 months_) were determined. Nutrient intakes were energy-adjusted by Nutrient Density and Residual methods. Correlation coefficients ranged from 0.18–0.58 for energy and energy-adjusted nutrient intakes, and the percentage of participants classified into the same quartile ranged from 28–47% between the FFQ_baseline_ and the 24-HDRs. For the FFQ_12 months_ compared with FFQ_baseline_, correlation coefficients ranged from 0.52–0.88 for intakes of energy, energy-adjusted nutrients, and food groups, and the proportion of participants classified into the same quartiles ranged from 43–69%. Overall, the FFQ provided a satisfactory ranking of individuals according to energy, nutrient, and food group intakes, making the FFQ suitable for use in epidemiological studies investigating diet in relation to disease outcomes.

## 1. Introduction

Dietary intake measured over a long period is the preferred exposure when investigating associations between diet and cancer risks due to the long latency of cancer development. A food frequency questionnaire (FFQ) is designed to capture an individual’s habitual diet; it is inexpensive and easy to administer on a large scale and is the most frequently used dietary assessment method in epidemiological studies. However, the FFQ relies on participant memory and is influenced by social desirability [1,2,3]. Additionally, dietary intake measured using an FFQ is prone to both random and systematic measurement errors. These measurement errors may lead to regression dilution bias, meaning that risk estimates of the association between dietary exposure and disease outcome will be attenuated toward the null [4]. An important step before initiating such studies is, therefore, to validate the FFQ, preferably with a reference method without overlapping measurement errors to avoid erroneous association estimations. Besides recovery biomarkers, weighted food records (WFR) are considered the “gold standard” within validation studies of dietary assessment methods, yet due to the heavy participant burden and risk of misreporting (change of diet), 24-h dietary recalls (24-HDR) are often used as a pragmatic alternative [5].

From 2015 to 2019, the Danish Diet, Cancer, and Health—Next Generations (DCH-NG) cohort was established to facilitate investigations of genetic, metabolomic, microbiomic, environmental, behavioral, and socioeconomic factors and their interactions in the development and prognosis of cancer as well as other non-communicable diseases. A validation sub-cohort called the DCH-NG MAX study was likewise established with the purpose, among others, of validating the FFQ administered in the DCH-NG cohort with 24-HDRs (myfood24) [6,7,8]. The web-based FFQ included 376 items. Web-based questionnaires have, in recent years, gained considerable attraction as they have several advantages over paper-based questionnaires: they are less burdensome for the participant due to easy access through a tablet or computer, and the possibility of skipping irrelevant questions can shorten the response time. For the researcher, the elimination of missing data, answer validation, ease of administration, as well as faster back-end nutrient analysis, are substantial improvements in terms of time and cost [9].

In order to evaluate the new web-based FFQ, the aims of the present study were: (i) to validate the FFQ with three 24-HDRs for energy and nutrient intakes and (ii) to assess the reproducibility of the FFQ over one year for energy, nutrient, and food group intakes in the DCH-NG MAX study.

## 2. Materials and Methods

### 2.1. Study Population

From August 2017 to February 2019, a sub-cohort (DCH-NG MAX) of the Danish DCH-NG cohort (*n* = 39,554) was established and included 720 participants. The purpose of DCH-NG MAX was to evaluate and validate dietary and lifestyle questionnaires, metabolomics, and microbiota, as well as to allow exploratory investigations of the associations between lifestyle, microbiota, genetics, and molecular phenotype. Participation in the DCH-NG MAX study included a clinical assessment and completion of 24-HDRs, an FFQ, and a lifestyle questionnaire (LSQ) at baseline, 6, and 12 months. At the clinical assessment, the following biospecimens were collected: blood, urine, saliva, and stool samples. In addition, height, weight, waist and hip circumference, total fat mass, fat-free mass, visceral fat, muscle mass, and blood pressure were measured by trained staff according to standardized protocols [6]. A timeline of data collection in the DCH-NG MAX sub-cohort is shown in Figure 1.

For analyses in the present study, inclusion criteria for validation analyses were a completed FFQ at baseline and three 24-HDRs (baseline, six, and twelve months) (*n* = 289). Participants with only weekend days for their 24-HDR were excluded (*n* = 12) as their intake was not considered to be representative of a habitual diet. For reproducibility analyses, inclusion criteria were completed FFQs at baseline and at 12 months (*n* = 415). A flowchart of the selection of DCH-NG MAX participants for the validity and reproducibility analyses is shown in Figure 2. To validate an FFQ, it has been suggested that a sample size of 200–300 subjects with three days of repeated measures per subject is required [10]. The sample sizes in the current study were therefore considered adequate for analyzing the validity and reproducibility of the intake of energy, nutrients, and food groups from the FFQ.

### 2.2. The FFQ

The FFQ is a web-based semi-quantitative FFQ designed to measure habitual dietary intake over the preceding 12 months with the purpose of allowing the ranking of individuals’ intake of nutrients and foods as well as assessing intake at group levels. The FFQ was based on previous FFQs: the paper-based FFQ from the DCH cohort and the web-based FFQ from the Danish National Birth cohort [11,12]. From 2014 to 2015, a thorough revision of the questionnaires was made. The revision involved modification and addition of questions along with updating and extending the food list, which resulted in a 376-item FFQ.

Participants were asked to report their frequency of consumption of foods and beverages during the previous 12 months, choosing between eight and twelve possible frequency categories ranging from never to eight times or more per day. For some foods and beverages, frequency of consumption was combined with a specified portion size, i.e., a slice of bread or a glass of water (thus semi-quantitative). Foods and beverages were grouped into the following categories: fermented milk products; cereals; bread; fat on bread; cold cuts; dishes; side dishes; cheese as a topping; sauces and dips; fats for cooking; sour cream and cream in sauces; raw vegetables; cooked vegetables; dressings; fruits and berries; preserved fruits; dried fruits; nuts, kernels and seeds; candy and chocolate; chips, pork crackling, and popcorn; cake and pastry; ice cream; desserts; condiments for desserts, cakes and fruits; dairy beverages; dairy replacement beverages; juices and lemonades; water, mineral water and soda; alcoholic beverages; coffee; tea; other drinks. Furthermore, participants were asked to report their use of dietary supplements as well as to respond to questions regarding dietary habits such as organic food consumption, eating out, or exclusion of major food groups.

All questions in the questionnaire were mandatory, and if a participant forgot to answer a question, a prompt would immediately appear. The questionnaire also incorporated skip functions, making it possible to skip several questions if a certain food group was not consumed. For instance, if cheese was not consumed, reporting “never” would allow the participant to skip further questions regarding cheese intake. Ancillary questions for specific items were added to the questionnaire to increase the precision of intake, such as the amount of butter on bread or the fat content and type of minced meat in dishes. The FFQ also contained cross-check questions for cold cuts; dishes; vegetables; fruits; desserts, ice cream, cakes, candies, chocolate, and snacks.

### 2.3. The 24-HDR (myfood24)

A web-based 24-HDR was used as the reference method for the validation of the FFQ. The 24-HDRs were collected using the Danish version of the online “myfood24” dietary assessment tool developed in Leeds, UK [7,8,13,14]. The Danish database for myfood24 was built by The Danish Cancer Society Research Center in 2017 and included a database of 1668 items covering both adult and infant foods. The Danish database was constructed through several steps. First, all food items from the Danish Food Composition Table (DFCT) (FRIDA Food, version 2, 2017) were manually examined, with the purpose of removing non-edible items. All edible items were added to the database [15]. Second, since the DCFT contains few cooked items, cooked varieties of all raw items were searched for in food composition tables (FCT) from Sweden (National Food Agency) and England (McCance and Widdowson’s version 6 and 7) [16,17,18]. Third, the FFQ was reviewed for commonly eaten dishes, and dishes not included in the database were found in either the Swedish or English FCTs and added to the database. Fourth, the branded myfood24 database was also scanned for typically consumed items [13]. Lastly, the database was checked by a nutritionist and clinical dietician for any forgotten items commonly consumed. In addition, images of portion sizes were incorporated and purchased from the Danish Food Institute containing 41 series with six different sizes per series from 2011 [19]. For items where images were not suitable, average portion sizes (small, medium, large) and household measurements (glass, tablespoon, etc.) were obtained from the report “Dimensions, weight and portion sizes of foods” from the Danish Food Institute from 2013 [20].

### 2.4. Administration of the FFQ and 24-HDR

The FFQ was designed for PCs using the online platform InMoment (South Jordan, UT, USA), though the questionnaire was also functional on tablets. Upon enrollment, participants received access to the baseline FFQ by logging on to a personal web profile. One month prior to the 6- and 12-month clinical assessments, participants received access to the respective FFQs. In total, three FFQs were administered: around baseline, 6, and 12 months. Reminders to complete the FFQs were sent out four times over a period of three months via e-mail and mobile text message. The 24-HDRs appeared on the personal web profile after completing the clinical assessment. The 24-HDRs were accessible until and including the day after the clinical assessment. A reminder text message was sent out the same day as the 24-HDRs became accessible, and a second reminder was sent out the following day. In total, six 24-HDRs were administered, two at each time point (baseline, 6 and 12 months). For the first 24-HDR, participants were instructed to register drinks and foods consumed the day prior to the clinical assessment (denoted as “yesterday”), and for the second 24-HDR, participants were instructed to register drinks and foods consumed on the day of the clinical assessment (denoted as “today”). Only the “yesterday” 24-HDRs were included in the current study because participants had various fasting times (0 to more than 9 h) prior to the clinical assessment, and therefore the “today” 24-HDRs were not considered representative of a typical day.

### 2.5. Analysis of Energy, Nutrient, and Food Intake from the FFQ and 24-HDR

FFQ data were transferred from the InMoment platform to the Danish Cancer Society Research Center for estimation of food, energy, and nutrient intake. Food intake (g/day) was estimated by multiplying the frequency of consumption of each food item with the corresponding standard portion size. Standard portion sizes were from EPIC-SOFT, The National Food Institute (DTU FOOD), and the Danish National Birth Cohort [12,20,21]. Subsequently, all food items from the FFQ were linked to food items in the Danish database, Frida Food Data, version 4, 2019 [22]. Energy (kJ/day), macronutrient (g/day), and micronutrient (mg/day or µg/day) intakes were further estimated using the software program FoodCalc version 1.3 [23]. Whole-grain intake was estimated based on the whole-grain content of the whole-grain products. Information about the whole-grain content was obtained from DTU FOOD. A detailed description of the estimation of whole-grain intake has been described in a previous study [24]. For the 24-HDR data food, energy and nutrient intakes were calculated directly in the online tool myfood24. Within myfood24, each food item was multiplied by the specified portion size and linked to its corresponding FCT: FRIDA food, Swedish National Food Agency, McCance and Widdowson’s, or myfood24 branded FCT [13,15,16,17,18].

### 2.6. Demographics, Anthropometrics, and Lifestyle Data

Sex and age were retrieved from the Danish Central Population Registry. Smoking habits, physical activity, and education were self-reported from the LSQ completed at baseline. Smoking habits were grouped as never, former, and current smokers. Physical activity, reported as sports, was calculated as hours per week [25]. Education was based on information about the highest attained education. Anthropometric measures such as weight (kg), BMI (kg/m^2^), and waist circumference (cm) were obtained from the clinical assessment at baseline and 12 months. BMI was classified into five groups: underweight (<18.5 kg/m^2^), normal weight (18.5–24.9 kg/m^2^), overweight (25–29.9 kg/m^2^), obese grade I (30–34.9 kg/m^2^) and obese grade II (≥35 kg/m^2^) [26]. Waist circumference was classified according to the risk of developing lifestyle diseases: no increased risk (men < 94 cm: women < 80 cm), increased risk (men 94–102 cm: women 80–88 cm), and substantially increased risk (men > 102 cm: women > 88 cm) [27].

### 2.7. Statistical Analyses

Descriptive statistics are presented as counts (%) or medians (P25–P75). Differences in median weight and waist circumference were assessed between the 12-month and baseline measurements. Energy adjustment is often applied in nutrition epidemiology to control for the confounding effect of energy in relation to a disease outcome [28]. We, therefore, adjusted nutrient intakes for the total caloric intake by considering (1) nutrient densities calculated by dividing the nutrient by the total caloric intake (2) residuals from a model of the nutrient linearly regressed on the total caloric intake (a double-log model was used for right-skewed outcomes) [28]. To improve normality, nutrient densities were log-transformed. The density model is a simple energy-adjustment approach; however, dividing the nutrient intake by total caloric intake does not remove the effect of energy fully. Therefore, we also applied the residual model [28]. Since dietary intake can vary from day to day and between weekends and weekdays as well as between seasons, we utilized the mean of three 24-HDRs to obtain habitual intake [29], which hereafter will be referred to as the 24-HDRs. To test for systematic differences in adjusted nutrient intakes between the FFQ_baseline_ and 24-HDRs, as well as between the FFQ_baseline_ and FFQ_12 months_, the Wilcoxon signed-rank test was used. Agreement between the FFQ_baseline_ and 24-HDRs, as well as between the FFQ_baseline_ and FFQ_12 months_, was assessed using the Bland–Altman (BA) method [30] and cross-classification. Due to the non-normality of most of the differences in adjusted intakes, 95% limits of agreement (LOAs) were determined non-parametrically using the upper and lower 2.5% percentiles of the differences [30,31]. LOAs are only reported for outcomes showing reasonable BA plots. For outcomes for which LOAs were assessed on the log scale, BA plots are illustrated on the original scale using back transformation as described in Euser et al. [32]. Cross-classification was performed for energy, nutrients, and food groups by dividing variables into quartiles and tabulating them against each other in order to assess the proportion of participants classified into the same, adjacent, opposite, and extreme opposite quartiles (the latter also referred to as misclassification). Lastly, correlations of intakes of energy, nutrients, and food groups between the FFQ_baseline_ and the 24-HDRs, as well as between the FFQ_baseline_ and FFQ_12 months_, were assessed using the Spearman correlation analysis due to non-normality of several nutrients and food groups [10]. Confidence intervals for the Spearman correlation were determined by the bootstrap method. Statistical analyses were carried out using R, version 4.1.2. Due to the large number of associations explored, only *p*-values below 0.01 were considered statistically significant.

## 3. Results

The baseline characteristics of the study participants for both the validation and reproducibility analyses are shown in Table 1. For both study populations, the median age of the participants was 50 years, and half had a BMI below 25 kg/m^2^ and a waist circumference < 94 cm for men and <80 cm for women. Approximately 60% of participants had a higher education of more than three years, and more than half had never smoked. Physical activity was 2.6 and 2.3 median hours per week for the participants in the validation analyses and the reproducibility analysis, respectively. There were no significant differences in weight and waist circumference during the study period. The 24-HDRs were distributed as follows: 76% weekdays and 24% weekend days, with 34% from spring/summer and 66% from autumn/winter. The median time to complete the FFQ (either baseline or 12-month FFQ) was 59 min and 18 min for the 24-HDRs. Participants with a completion time > 24 h, stemming from the fact that participants had the possibility to complete the FFQ In separate sessions, were excluded from the time calculations.

### 3.1. Relative Validity of Energy and Nutrient Intakes

Median energy and nutrient intakes from the FFQ_baseline_ and the 24-HDRs are presented in Table 2. Reported absolute intakes of energy and all nutrients were higher with the FFQ_baseline_ compared to the 24-HDRs.

Significant differences in median intakes of energy and most nutrient densities were found between the FFQ_baseline_ and the 24-HDRs, except for monounsaturated fatty acids (MUFA), carbohydrates, alcohol, and sodium densities (Table 3). No differences were found for residual nutrient intakes between the FFQ_baseline_ and the 24-HDRs, except for vitamin B6 (Appendix A). The Spearman correlation coefficient for energy intake between the FFQ_baseline_ and the 24-HDRs was 0.26 (95% CI: 0.15–0.36). For nutrient densities and nutrient residuals, the correlation coefficients ranged from 0.18 to 0.58 (Table 3 and Appendix A). Bland–Altman plots illustrating the difference in nutrient intakes between the FFQ_baseline_ compared with the 24-HDRs are given in Figure 3 and Appendix A. LOAs were estimated for intakes of energy, fat, saturated fatty acids (SFA), MUFA, polyunsaturated fatty acids (PUFA), protein, carbohydrate, dietary fiber, calcium, phosphorous, and riboflavin densities (Table 3) but were not possible to determine for the rest of the nutrient densities, illustrated for iron and vitamin C densities in Figure 3E,F. Similar results were observed for nutrient residuals (Appendix A). The proportion of participants classified into the same quartile for energy and nutrient densities estimated from FFQ and 24-HDR ranged from 28% to 45%. Misclassification (extreme opposite quartiles) was less than 10% for energy and all nutrient density intakes (Table 3). The proportion of participants classified into the same quartile for nutrient residuals ranged from 29% to 47%, and overall misclassification was likewise below 10% (Appendix A).

### 3.2. Reproducibility of Energy, Nutrient, and Food Group Intakes

Median energy, nutrient, and food group intakes from the FFQ_baseline_ and the FFQ_12 months_ are presented in Table 4 and Table 5. Reported absolute intakes of energy, all nutrients, and food groups were lower for the FFQ_12 months_ compared to the FFQ_baseline_, except for legumes and soft drinks, where the intakes from the FFQ_12 months_ were marginally higher compared with the FFQ_baseline_.

There were significant differences between the FFQ_12 months_ and the FFQ_baseline_ for median intakes of energy as well as cholesterol, carbohydrate, dietary fiber, alcohol, magnesium, copper, selenium, retinol, and vitamin D densities (Table 6). Conversely, there were no significant differences in any of the nutrient residuals between the FFQ_12 months_ and the FFQ_baseline_ (Appendix A). Most of the food group intakes showed significant differences between the FFQ_12 months_ and FFQ_baseline_, except for intakes of legumes, fast food, and soft drinks (Table 7). The Spearman correlation coefficient for energy intake between the FFQ_12 months_ and the FFQ_baseline_ was 0.67 (95% CI: 0.61–0.73). For nutrient densities and nutrient residuals, the correlation coefficients ranged from 0.52 to 0.80 (Table 6 and Appendix A). For food groups, the correlation coefficients ranged from 0.60 to 0.88 (Table 7). Bland–Altman plots illustrating the differences in intakes between the FFQ_12 months_ and the FFQ_baseline_ are given in Figure 4 and Appendix A. LOAs were estimated for intakes of energy and most nutrient densities (Table 6). LOAs were not possible to assess for eicosapentaenoic acid (EPA), docosahexaenoic acid (DHA), cholesterol, alcohol, selenium, iodine, retinol, vitamin D, vitamin B6, and vitamin B12 densities. In addition, similar results were observed for nutrient residuals (Appendix A). For food group intakes, LOA was calculated for those that did not contain zero intakes: fruits, vegetables, potatoes, whole grains, eggs, dairy products, fermented dairy products, and fat. The proportion of participants classified into the same quartile for energy, nutrient densities, and nutrient residuals in both FFQs ranged from 43% to 56%, and misclassification was less than five percent (Table 6 and Appendix A). Lastly, the proportion of participants classified into the same quartile for the food groups ranged between 45% to 69%, with a misclassification below four percent (Table 7).

## 4. Discussion

The purpose of the current study was to validate the FFQ used in the DCH-NG cohort with 24-HDRs. Overall, participants reported higher absolute energy, macronutrient, and micronutrient intakes assessed with the FFQ compared with the 24-HDRs. In addition, we found differences in intakes for most nutrient densities. Differences in residuals were not expected as the concept of the residual model includes that the sum of residuals is equal to zero.

Higher reported absolute intakes of energy and macronutrients using a web-based FFQ compared to 24-HDRs, food diaries, or WFRs have been described in several validation studies [33,34,35,36,37,38,39]. However, other studies have also reported the opposite, that higher absolute intakes were reported with the 24-HDRs or WFRs compared to FFQs [40,41,42,43]. According to the Nordic Nutrition Recommendations, the daily recommended energy intake ranges from 8100–13,200 kJ depending on age, sex, and physical activity level [44]. Thus, comparing the recommended energy requirements with the median energy intake estimated in the current study from the FFQ (10,832 kJ) and 24-HDRs (8491 kJ), we speculate that the intakes estimated with 24-HDRs could be low, considering that our population includes both men and women from the age of 18 to 67 years. To adequately estimate energy and nutrient intakes using 24-HDRs or WFR, it has been advised to collect these for seven days or more [45]. The number of 24-HDRs in the current study was lower (three 24-HDRs). Furthermore, we were not able to assess agreement (Bland–Altman method and cross-classification) of food group intakes between the FFQ and the 24-HDRs due to a high proportion of zero-intakes in the 24-HDRs. Together, this could imply that the number of recording days was too low to reflect habitual dietary intake, including missing out on weekend days where the energy intake possibly is higher. On the contrary, the energy intake, as well as macronutrient intakes estimated with the FFQ, appears high in the current study in comparison with intakes from the Danish National Survey from 2011–2013, which are based on seven-day food records (median absolute intake: energy 9400 kJ, total fat 92 g, protein 85 g, carbohydrates, 230 g) [46]. In addition, a large review of national dietary surveys, including data from 21 European countries, also reported lower energy intake compared with that estimated from the FFQ from the current study [47]. A Norwegian study by Medin et al. further estimated total energy expenditure (TEE) using a doubly labeled water technique (DLW), on a sub-sample of 30 women, as a true measure of energy intake. The authors then observed that estimated energy intake was, on average, six percent lower for the FFQ and 17% lower for the mean of 4 × 24-HDRs and proposed that the difference in intake estimation could possibly be due to underestimation by the mean of 4 × 24-HDRs [33]. However, this notion is in contrast to an older but larger study by Freedman et al. that pooled results from five validation studies comparing energy intake of FFQs and single 24-HDRs with that of TEE. The authors observed 13% lower energy intakes from the single 24-HDRs and 31% from FFQs compared with TEE for men. For women, corresponding figures were 18% lower in energy intakes from the single 24-HDRs and 28% from FFQs [48]. Even though the DLW method is referred to as the “gold standard”, underlying calculations for the estimation of TEE have been reviewed and updated from time to time. Therefore, there may also be differences in how the TEE has been estimated between studies [49,50].

We further suggest some methodological explanations for the discrepancy observed in the current study between intakes reported with the FFQ compared with the 24-HDR. First, long lists of food items in an FFQ (>100 items) may lead to overestimation [10], and since the current FFQ contains 376 items, this could be one of the reasons why we see a higher intake estimated with the FFQ compared with the 24-HDRs. Second, the recall process of the two methods differs. When filling out an FFQ, participants are asked to report frequencies and quantities of a predefined food list of generally consumed items. This is in contrast with the open-ended 24-HDR, with no predefined food list and, therefore, no assistance in the memory of what the participant has eaten. This could lead to lower reported intake with the 24-HDR, simply due to forgotten foods and beverages. Lastly, differences in food portions between the FFQ (with standardized food portion sizes) and the 24-HDRs (self-reported food portion sizes based on a series of food images or household measures) could also influence the differences in estimated intakes.

In the current study, correlation coefficients of energy, nutrient densities, and nutrient residuals ranged from 0.18 to 0.58 between the FFQ_baseline_ and the 24-HDRs. Other studies validating web-based FFQs with either 24-HDRs or WFR (with four to twelve recording days) have reported both lower and higher energy-adjusted nutrient correlations ranging from close to zero to almost 0.9 [33,40,42,51]. However, the latter studies are population dependent (i.e., they differ in population size, age range, sex distribution, nationality, etc.), effectively hindering a direct comparison of correlation strengths. Moreover, caution should be taken when interpreting correlations in validation studies since intakes with high correlations can still be subject to systematic errors, and, most importantly, high correlations are not equated with having good agreement [52,53,54].

To estimate agreement between the FFQ_baseline_ and the 24-HDRs, we used the Bland–Altman method [31] and cross-classification. The Bland–Altman plot illustrates the mean of the two methods on the x-axis and the difference between the two methods on the y-axis, which enable us to uncover systematic differences [31]. Since the FFQ is designed to capture long-term dietary intake with the purpose of ranking individuals’ intakes [55] in contrast to the 24-HDR, which is a tool where absolute dietary intake can be assessed over a short period [29], we did not expect complete agreement between the two methods. In addition, Cade et al. also stated in a review regarding the development, validation, and utilization of FFQs, that making cut-offs for what could be considered acceptable Bland–Altman LOAs for intakes of energy, nutrients, and food groups is not meaningful since this would depend on the objectives of a specific study [5]. With that in mind, we were able to calculate LOAs for intakes of energy and some nutrient densities, and nutrient residuals. The Bland–Altman plots for the back-transformed energy and nutrient densities showed a bias with an increased spread of the differences (LOA) with increased mean of the outcomes, which explained the funnel shape in the plots. However, on the log scale, the median bias and LOA were constant. Furthermore, the Bland–Altman plots for both nutrient densities and nutrient residuals (EPA, DHA, cholesterol, total sugar, alcohol, and most micronutrients) indicated a bias such that higher intakes were reported with the FFQ than the 24-HDRs at low intakes and, conversely the FFQ reported lower intakes than the 24-HDRs for high intakes. This bias was, for instance, illustrated in the Bland–Altman plots for iron and vitamin C residuals in the current study. Part of this could be explained by inherent errors in the FFQ, such as overreporting of low intakes due to many food items within a food group. The lower intake reported by the FFQ for high intakes may also be due to incorporated frequency restrictions in the FFQ, where for instance, the highest reported frequency for each fruit item was “two or more times per day.”

However, the cross-classification analysis showed that nutrient intakes (both nutrient densities and residuals) were reasonably accurate in ranking individuals comparing the FFQ with 24-HDRs. Participants classified into the same quartile ranged from 28–47%, whereas participants classified into the same or adjacent quartile ranged from 69–86%. In addition, misclassification was below 10% for both nutrient densities and nutrient residuals. Other studies evaluating the ranking ability of an FFQ with that of 24-HDRs or WFR have shown similar results: where participants classified into the same quartile ranged from 20–70%, and participants classified into the same or adjacent quartile ranged from 50–95% [33,35,36,40,56].

We further assessed the reproducibility of the FFQ with a one-year administration interval. Overall, reported intakes of energy, nutrients, and most of the food groups were lower at 12 months compared to baseline. We further observed differences in intakes of energy, cholesterol, carbohydrate, dietary fiber, alcohol, magnesium, copper, selenium, retinol, and vitamin D densities, as well as most of the food groups, but not for nutrient residuals between the FFQ_12 months_ and the FFQ_baseline_. Lower estimated intakes with the second administration of an FFQ have been reported by previous studies with both short (more than three months) [35,36,39,51,57] and long administration intervals (one year) [43,56]. Some studies have suggested a learning effect between the first and second administration of the FFQs [39,43,57]. Moreover, a long recall period (one year) could also influence reproducibility due to dietary changes. Yet, we did not find significant changes in either weight or waist circumference of the participants in the current study, which could indicate that the participants did not change their diets (or at least any change did not affect their energy balances) substantially. High correlation coefficients were observed for all nutrient density and nutrient residual intakes (0.52 to 0.80) as well as food group intakes (0.60 to 0.88) comparing the FFQ_baseline_ with the FFQ_12 months_. The correlation coefficients, as well as the proportion of participants classified into the same and/or adjacent quartiles of dietary intakes in the current study, were in similar ranges as reported in other studies for FFQs with both short (more than three months) and long (one year) recall periods [35,36,39,43,51,57].

### Strengths and Limitations

The current study has a relatively large sample size (*n* = 300–400) compared with other FFQ validity and reproducibility studies, where the majority of the studies have a sample size below 200 [33,34,35,36,39,40,42,43,51,56]. Additionally, we do not assume to have a seasonal bias since the FFQ covers a whole year, and participants were enrolled from August until the end of February; hence the 24-HDRs also cover different seasons. However, we did have a lower proportion of 24-HDRs completed during summer/spring (34%) compared to autumn/winter (66%). The current study also has limitations. We were not able to compare food group intakes of the FFQ_baseline_ with the 24-HDRs, due to a large proportion of zero-intakes reported in the 24-HDRs. This effectively highlights that three 24-HDRs are not enough to capture foods and beverages that are not consumed regularly, even though both weekdays and weekend days were recorded. When validating an FFQ, it is important to use a reference method with no or limited overlapping measurement errors. Similar measurement errors can result in artificially high correlations compared with the true measurement [58]. Therefore, we cannot rule out that common errors, such as the recall of foods eaten and social desirability, could influence the correlations [1,2,3]. Lastly, the period for the reported dietary intake differs between the instruments. The FFQ_baseline_ refers to foods and beverages consumed in the past year (looking back in time) in contrast to the 24-HDRs, which refers to foods and beverages consumed at specific days at baseline, six, and twelve months. However, the justification for choosing the FFQ_baseline_ and not the FFQ_12 months_ for validation is because the FFQ is only completed at baseline in the main cohort (DCH-NG).

## 5. Conclusions

Overall, we conclude that reported intakes using the FFQ are suitable for use in disease end-point studies with diet as the exposure. The FFQ_basline_ provided a satisfactory ranking of participants into the same or adjacent quartile relative to the 24-HDRs and the FFQ_12 months_. However, we observed higher absolute energy and nutrient intakes reported with the FFQ compared with the 24-HDRs. Differences in reported intakes were seen for nutrient densities but not for the nutrient residuals. In addition, we observed lower absolute intakes of energy, nutrients, and most food groups for the FFQ_12 months_ compared with the FFQ_baseline_. The difference in reported intakes was observed for most food groups and some of the nutrient densities but again not for the nutrient residuals. The validity and reproducibility of the DCH-NG FFQ are in comparison with similar validation and reproducibility studies.

## Figures and Tables

**Figure 1 nutrients-15-02389-f001:**
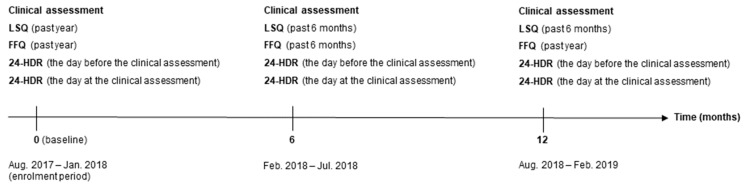
Timeline for the data collection in the Diet, Cancer, and Health—Next Generations (DCH-NG) MAX study including clinical assessment and completion of lifestyle questionnaire (LSQ), food frequency questionnaire (FFQ), and 24-h dietary recall (24-HDR).

**Figure 2 nutrients-15-02389-f002:**
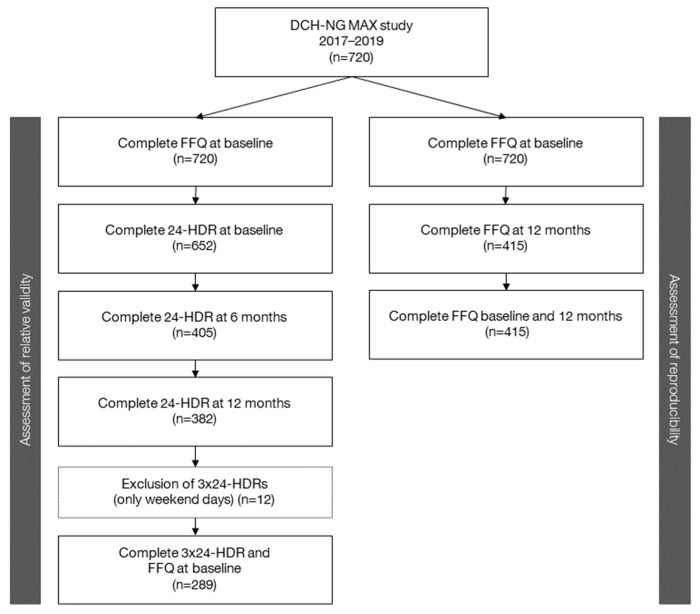
Flowchart of participants for assessment of validity and reproducibility.

**Figure 3 nutrients-15-02389-f003:**
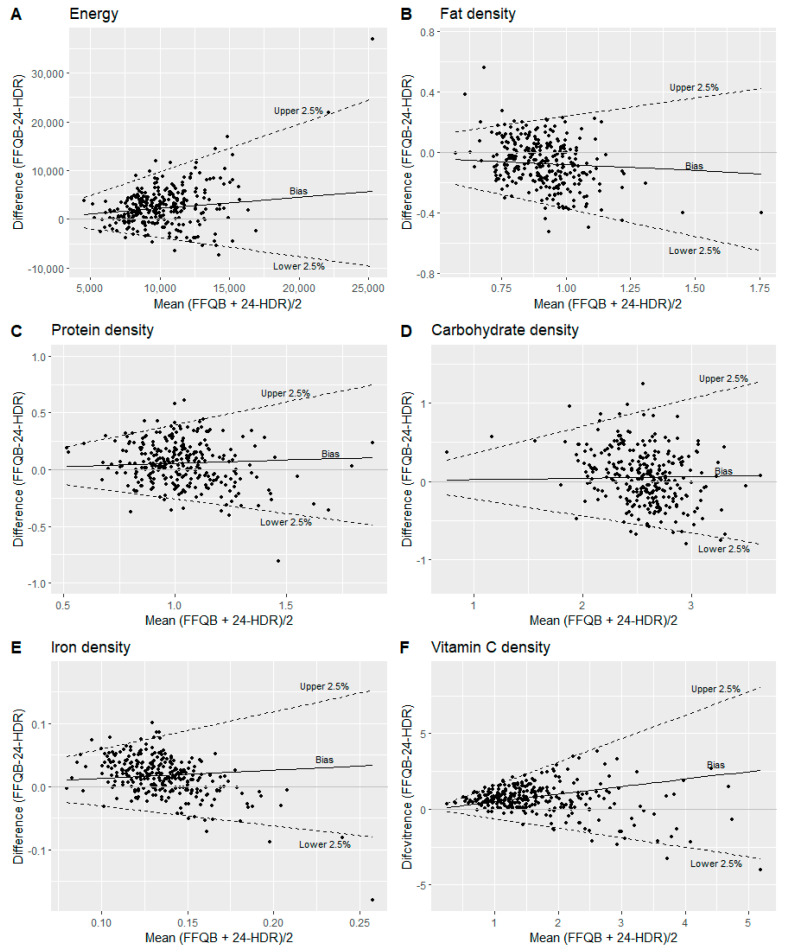
Bland–Altman plots comparing the FFQ_baseline_ with the 24-HDRs for (**A**) energy (kJ); (**B**) fat density (g/kJ); (**C**) protein density (g/kJ); (**D**) carbohydrate density (g/kJ); (**E**) iron density (mg/kJ); (**F**) vitamin C density (mg/kJ). The solid line illustrates the median difference, and the dotted lines illustrate the upper and lower 2.5%.

**Figure 4 nutrients-15-02389-f004:**
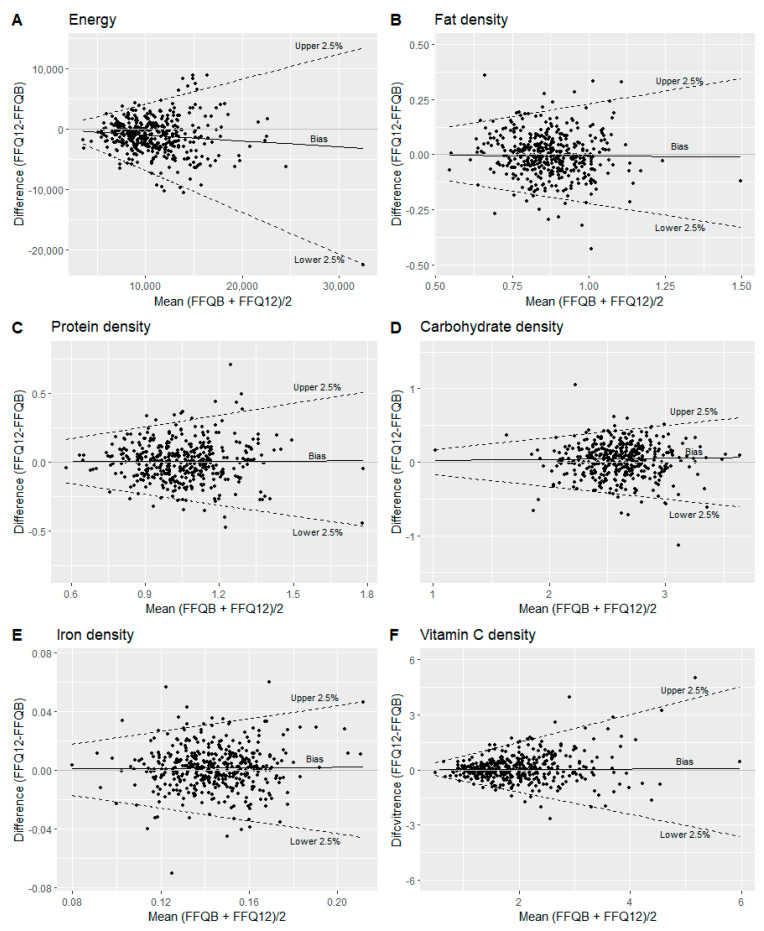
Bland–Altman plots comparing the FFQ_baseline_ with the FFQ_12 months_ for (**A**) energy (kJ); (**B**) fat density (g/kJ); (**C**) protein density (g/kJ); (**D**) carbohydrate density (g/kJ); (**E**) iron density (mg/kJ); (**F**) vitamin C density (mg/kJ). The solid line illustrates the median difference, and the dotted lines illustrate the upper and lower 2.5%.

**Table 1 nutrients-15-02389-t001:** Baseline characteristics of DCH-NG MAX participants included in the relative validity and reproducibility analysis, respectively.

	Relative Validity			Reproducibility		
	All	Men	Women	All	Men	Women
	*n* = 289	*n* = 130	*n* = 159	*n* = 415	*n* = 188	*n* = 227
	*n* (%)	*n* (%)	*n* (%)	*n* (%)	*n* (%)	*n* (%)
**Age (years)**						
Median (p25–p75)	50 (40–54)	50 (44–55)	49 (32–53)	50 (40–54)	51 (43–55)	49 (32–53)
18–34	66 (23)	21 (16)	45 (28)	95 (23)	33 (18)	62 (27)
35–50	102 (35)	47 (36)	55 (35)	140 (34)	61 (32)	79 (35)
>50	121 (42)	62 (48)	59 (37)	180 (43)	94 (50)	86 (38)
**Weight (kg)**						
Median (p25–p75)	74 (64–83)	82 (77–91)	67 (61–74)	75 (64–84)	82 (77–92)	67 (61–74)
**BMI (kg/m^2^)**						
Underweight (<18.5)	4 (1)	0 (0)	4 (3)	3 (1)	0 (0)	3 (1)
Normal weight (18.5–24.9)	155 (54)	59 (45)	96 (60)	217 (52)	77 (41)	140 (62)
Overweight (25–29.9)	102 (35)	57 (44)	45 (28)	155 (37)	93 (50)	62 (27)
Obese (≥30)	28 (10)	14 (11)	14 (9)	40 (10)	18 (10)	22 (10)
**Waist circumference**						
No increased risk (men < 94 cm; women < 80 cm)		73 (56)	72 (45)		96 (51)	97 (43)
Increased risk (men 94–102 cm; women 80–88 cm)		34 (26)	46 (29)		57 (30)	70 (31)
Substantially increased risk (men > 102 cm; women > 88 cm)		23 (18)	41 (26)		35 (19)	60 (26)
**Highest attained education**						
Higher education (>4 years)	101 (35)	46 (35)	55 (35)	137 (33)	62 (33)	75 (33)
Higher education (3–4 years)	86 (30)	35 (27)	51 (32)	113 (27)	44 (23)	69 (30)
Higher education (1–2 years)	31 (11)	18 (14)	13 (8)	48 (12)	24 (13)	24 (11)
Vocational training	63 (22)	27 (21)	36 (23)	101 (24)	50 (27)	51 (23)
Basic school	8 (3)	4 (3)	4 (3)	16 (4)	8 (4)	8 (4)
**Smoking habits**						
Never	163 (56)	76 (59)	87 (55)	224 (54)	106 (56)	118 (52)
Former	75 (26)	26 (20)	49 (31)	112 (27)	41 (22)	71 (31)
Current	51 (18)	28 (22)	23 (15)	79 (19)	41 (22)	38 (17)
**Physical activity (sports)**						
Median hours per week (p25–p75)	2.6 (0.8–4.3)	2.6 (1.1–4.3)	2.2 (0.6–4.3)	2.3 (0.7–4.1)	2.6 (0.9–4.2)	2.2 (0.6–3.8)

**Table 2 nutrients-15-02389-t002:** Median daily (p25–p75) absolute intakes of energy and nutrients from FFQ_baseline_ and the 24-HDRs (*n* = 289).

Energy and Nutrients	FFQ Baseline		24-HDRs	
	Median	(p25–p75)	Median	(p25–p75)
Energy, kJ	10,832	(9136–13,269)	8491	(7200–10,099)
Protein, g	114	(94–139)	85	(69–103)
Total fat, g	96	(77–118)	82	(64–100)
SFA, g	30	(24–40)	26	(20–33)
MUFA, g	35	(28–44)	29	(22–36)
PUFA, g	15	(12–19)	14	(10–18)
EPA, g	0.16	(0.09–0.27)	0.03	(0.00–0.16)
DHA, g	0.25	(0.15–0.40)	0.08	(0.01–0.27)
Cholesterol, mg	372	(286–475)	217	(148–303)
Carbohydrate, g	281	(227–345)	215	(178–263)
Total sugar, g	122	(91–156)	62	(48–88)
Fibre, g	33	(25–42)	21	(16–27)
Alcohol, g	10	(3.7–18)	5.5	(0.00–15)
Sodium, mg	3582	(2901–4417)	2815	(2239–3635)
Potassium, mg	5096	(4222–6111)	3068	(2569–3673)
Calcium, mg	1444	(1097–1895)	877	(692–1090)
Magnesium, mg	520	(421–626)	339	(276–422)
Phosphorus, mg	2067	(1687–2574)	1313	(1078–1604)
Iron, mg	15	(13–19)	11	(8.4–13)
Copper, mg	5.0	(3.7–6.4)	3.5	(2.3–4.8)
Zink, mg	15	(13–19)	10	(8.0–13)
Selenium, µg	68	(52–84)	43	(31–55)
Iodine, µg	267	(202–347)	108	(77–155)
Retinol, µg	448	(295–733)	250	(157–486)
Beta carotene, µg	7570	(4742–11,011)	2578	(801–5493)
Vitamin A, µg	1218	(901–1640)	740	(439–1129)
Vitamin D, µg	4.6	(3.4–6.3)	2.3	(1.3–4.1)
Vitamin E, mg	12	(10–16)	8.0	(5.8–11)
Vitamin K, µg	245	(171–342)	37	(13–87)
Thiamine, mg	1.7	(1.4–2.1)	1.2	(1.0–1.6)
Riboflavin, mg	2.4	(1.9–3.1)	1.4	(1.1–1.7)
Niacin, mg	22	(18–27)	16	(12–21)
Vitamin B6, mg	2.4	(1.9–2.8)	1.5	(1.1–1.9)
Vitamin B12, µg	7.9	(5.9–10)	3.9	(2.5–5.9)
Vitamin C, mg	207	(148–279)	88	(54–153)
Folate, µg	515	(407–645)	271	(190–356)

**Table 3 nutrients-15-02389-t003:** Energy and density-adjusted nutrient intakes from the FFQ_baseline_ compared with the 24-HDRs for median bias, Bland–Altman LOA, Spearman’s correlation coefficient, and cross-classification (*n* = 289).

Energy and Density Adjusted Nutrients ^a^	Median Bias ^c^	*p*-Value ^d^	Bland–Altman LOA ^e^		Spearman’s Correlation Coefficient	Cross-Classification			
			Lower	Upper	(95% CI)	Same Quartile (%)	Adjacent Quartile (%)	Opposite Quartile (%)	Extreme Opposite Quartile(%)
Energy, kJ	25	<0.0001	−32	189	0.26 (0.15–0.36)	30	40	24	6
Protein, g/kJ	1.06	<0.0001	−23	49	0.50 (0.40–0.58)	40	41	15	4
Total fat, g/kJ	0.92	<0.0001	−31	27	0.43 (0.34–0.52)	35	44	17	5
SFA, g/kJ	0.95	0.0006	−40	57	0.46 (0.36–0.55)	39	38	20	3
MUFA, g/kJ	0.98	0.0678	−38	63	0.37 (0.25–0.47)	35	40	19	7
PUFA, g/kJ	0.87	<0.0001	−51	61	0.37 (0.26–0.47)	31	46	18	6
EPA, g/kJ ^b^	0	<0.0001	- ^f^	- ^f^	0.20 (0.07–0.31)	28	42	22	8
DHA, g/kJ ^b^	0	<0.0001	- ^f^	- ^f^	0.28 (0.16–0.39)	29	43	20	7
Cholesterol, g/kJ	1.34	<0.0001	- ^f^	- ^f^	0.37 (0.26–0.48)	37	38	20	6
Carbohydrate, g/kJ	1.02	0.0037	−20	43	0.50 (0.39–0.59)	39	40	18	3
Total sugar, g/kJ	1.50	<0.0001	- ^f^	- ^f^	0.51 (0.41–0.60)	42	38	16	4
Fibre, g/kJ	1.21	<0.0001	−31	112	0.57 (0.48–0.65)	42	42	13	3
Alcohol, g/kJ ^b^	0.01	0.9708	- ^f^	- ^f^	0.53 (0.44–0.62)	- ^g^	- ^g^	- ^g^	- ^g^
Sodium, mg/kJ	0.97	0.7597	- ^f^	- ^f^	0.28 (0.17–0.38)	32	40	21	7
Potassium, mg/kJ	1.28	<0.0001	- ^f^	- ^f^	0.47 (0.37–0.56)	34	46	15	5
Calcium, mg/kJ	1.29	<0.0001	−33	148	0.38 (0.27–0.48)	36	41	17	6
Magnesium, mg/kJ	1.17	<0.0001	-^f^	-^f^	0.57 (0.48–0.64)	43	41	14	2
Phosphorus, mg/kJ	1.22	<0.0001	−18	89	0.38 (0.26–0.48)	37	38	19	7
Iron, mg/kJ	1.14	<0.0001	- ^f^	- ^f^	0.40 (0.30–0.50)	36	39	19	6
Copper, mg/kJ	1.09	<0.0001	- ^f^	- ^f^	0.58 (0.48–0.65)	45	41	11	3
Zink, mg/kJ	1.19	<0.0001	- ^f^	- ^f^	0.29 (0.17–0.39)	31	39	21	8
Selenium, µg/kJ	1.24	<0.0001	- ^f^	- ^f^	0.35 (0.23–0.45)	33	39	22	6
Iodine, µg/kJ	1.87	<0.0001	- ^f^	- ^f^	0.27 (0.14–0.39)	35	34	23	7
Retinol, µg/kJ	1.32	<0.0001	- ^f^	- ^f^	0.43 (0.32–0.52)	37	43	16	5
Beta carotene, µg/kJ	2.24	<0.0001	- ^f^	- ^f^	0.40 (0.30–0.51)	38	42	14	6
Vitamin A, µg/kJ	1.29	<0.0001	- ^f^	- ^f^	0.31 (0.18–0.41)	33	39	20	7
Vitamin D, µg/kJ	1.56	<0.0001	- ^f^	- ^f^	0.27 (0.16–0.38)	34	35	24	7
Vitamin E, mg/kJ	1.24	<0.0001	- ^f^	- ^f^	0.52 (0.43–0.60)	42	41	15	2
Vitamin K, µg/kJ ^b^	1.54	<0.0001	- ^f^	- ^f^	0.37 (0.26–0.47)	34	40	21	5
Thiamine, mg/kJ	1.06	0.0011	- ^f^	- ^f^	0.31 (0.20–0.41)	37	38	18	7
Riboflavin, mg/kJ	1.36	<0.0001	−17	154	0.44 (0.34–0.53)	39	38	19	4
Niacin, mg/kJ	1.05	0.0005	- ^f^	- ^f^	0.38 (0.27–0.48)	35	43	16	6
Vitamin B6, mg/kJ	1.26	<0.0001	- ^f^	- ^f^	0.36 (0.26–0.46)	35	41	19	6
Vitamin B12, µg/kJ	1.59	<0.0001	- ^f^	- ^f^	0.34 (0.23–0.44)	36	40	17	8
Vitamin C, mg/kJ	1.66	<0.0001	- ^f^	- ^f^	0.46 (0.35–0.56)	37	43	16	4
Folate, µg/kJ	1.51	<0.0001	- ^f^	- ^f^	0.48 (0.38–0.57)	40	40	16	4

^a^ Based on log-transformed density intakes. ^b^ Based on raw intake density intakes. ^c^ Median bias is reported as a percentage for log-transformed density intakes and unit difference for raw density intakes. ^d^ *p*-value, the test of difference in intake between (log^a^) FFQ_baseline_ and (log^a^) mean of three 24-HDRs using Wilcoxon signed-rank test. ^e^ Bland–Altman limits of agreement (LOAs) are reported as a percentage difference. ^f^ Bland–Altman limits of agreement (LOAs) are not reported as LOAs depend on the level of the nutrient. ^g^ Alcohol had a large proportion of non-consumers.

**Table 4 nutrients-15-02389-t004:** Median daily (p25–p75) absolute intakes of energy and nutrients from FFQ_baseline_ and FFQ_12 months_ (*n* = 415).

Energy and Nutrients	FFQ Baseline		FFQ 12 Months	
	Median	(p25–p75)	Median	(p25–p75)
Energy, kJ	10,884	(9043–13,411)	9827	(7974–12,052)
Protein, g	114	(95–142)	105	(82–129)
Total fat, g	95	(76–120)	86	(67–108)
SFA, g	30	(24–40)	27	(21–36)
MUFA, g	35	(27–45)	32	(25–40)
PUFA, g	15	(12–19)	14	(11–18)
EPA, g	0.17	(0.10–0.28)	0.15	(0.08–0.25)
DHA, g	0.27	(0.15–0.42)	0.23	(0.13–0.37)
Cholesterol, mg	373	(287–479)	322	(239–425)
Carbohydrate, g	282	(227–356)	260	(204–318)
Total sugar, g	122	(92–155)	107	(82–142)
Fibre, g	33	(25–42)	31	(22–40)
Alcohol, g	9.1	(3.6–18)	7.1	(3.0–14)
Sodium, mg	3615	(2912–4493)	3296	(2651–4138)
Potassium, mg	5096	(4218–6146)	4618	(3707–5723)
Calcium, mg	1444	(1099–1922)	1317	(1014–1768)
Magnesium, mg	512	(418–635)	467	(379–595)
Phosphorus, mg	2082	(1685–2621)	1895	(1516–2349)
Iron, mg	16	(12–19)	14	(12–17)
Copper, mg	4.9	(3.6–6.4)	4.7	(3.4–6.1)
Zink, mg	15	(13–19)	14	(11–17)
Selenium, µg	68	(53–89)	60	(46–78)
Iodine, µg	270	(207–363)	257	(196–332)
Retinol, µg	443	(294–756)	399	(260–663)
Beta carotene, µg	7310	(4639–10,948)	6556	(3990–10,399)
Vitamin A, µg	1206	(827–1654)	1060	(765–1474)
Vitamin D, µg	4.7	(3.4–6.4)	4.1	(2.8–6.0)
Vitamin E, mg	13	(10–16)	12	(8.4–15)
Vitamin K, µg	244	(167–345)	232	(154–343)
Thiamine, mg	1.7	(1.4–2.1)	1.5	(1.2–1.9)
Riboflavin, mg	2.4	(1.9–3.1)	2.2	(1.7–2.8)
Niacin, mg	22	(18–27)	20	(15–25)
Vitamin B6, mg	2.4	(1.9–2.9)	2.1	(1.7–2.6)
Vitamin B12, µg	8.2	(6.0–10.6)	7.1	(5.1–9.6)
Vitamin C, mg	207	(149–277)	190	(130–262)
Folate, µg	515	(403–647)	472	(362–609)

**Table 5 nutrients-15-02389-t005:** Median daily (p25–p75) absolute intakes of food groups from FFQ_baseline_ and FFQ_12 months_ (*n* = 415).

Food Groups	FFQ Baseline		FFQ 12 Months	
	Median	(p25–p75)	Median	(p25–p75)
Fruits, g	171	(103–256)	143	(90–229)
Vegetables, g	441	(323–591)	402	(294–561)
Potatoes, g	67	(42–108)	61	(39–96)
Legumes, g	4.6	(2.4–9.3)	4.7	(2.2–10)
Eggs, g	25	(17–38)	22	(15–33)
Poultry, g	32	(19–52)	30	(18–45)
Red meat, g	74	(45–110)	64	(38–91)
Processed red meat, g	20	(11–32)	16	(8–28)
Fast food, g	39	(21–59)	36	(21–58)
Fish and seafood, g	42	(26–65)	37	(21–59)
Dairy products, g	363	(173–646)	324	(130–617)
Fermented dairy products, g	116	(63–208)	95	(53–184)
Fat, g	22	(16–29)	20	(14–26)
Soft drinks, g	52	(18–182)	53	(17–168)
Coffee, g	504	(189–806)	448	(177–784)
Tea, g	89	(23–376)	67	(17–243)
Whole grain, g ^a^	62	(37–98)	57	(34–89)
Fruits, g	171	(103–256)	143	(90–229)
Vegetables, g	441	(323–591)	402	(294–561)
Potatoes, g	67	(42–108)	61	(39–96)

^a^ Estimated whole-grain intake from whole-grain products.

**Table 6 nutrients-15-02389-t006:** Energy and density-adjusted nutrient intakes from the FFQ_baseline_ compared with the FFQ_12 months_ for median bias, Bland–Altman LOAs, Spearman’s correlation coefficient, and cross-classification (*n* = 415).

Energy and Density Adjusted Nutrients ^a^	Median Bias ^c^	*p*-Value ^d^	Bland–Altman LOA ^e^		Spearman’s Correlation Coefficient	Cross-Classification			
			Lower	Upper	(95% CI)	Same Quartile (%)	Adjacent Quartile (%)	Opposite Quartile (%)	Extreme Opposite Quartile(%)
Energy, kJ	−13	<0.0001	−64	83	0.67 (0.61–0.73)	45	43	10	1
Protein, g/kJ	−0.5	0.8340	−25	30	0.66 (0.60–0.72)	47	40	10	3
Total fat, g/kJ	0.7	0.5480	-21	25	0.65 (0.57–0.71)	50	36	12	2
SFA, g/kJ	1.2	0.0160	−29	42	0.68 (0.62–0.73)	49	40	9	1
MUFA, g/kJ	−0.1	0.7370	−25	35	0.61 (0.54–0.67)	45	39	14	2
PUFA, g/kJ	−0.7	0.1450	−32	45	0.67 (0.60–0.72)	47	41	10	1
EPA, g/kJ ^b^	0	0.0630	- ^f^	- ^f^	0.68 (0.60–0.74)	52	36	10	2
DHA, g/kJ ^b^	0	0.0300	- ^f^	- ^f^	0.69 (0.63–0.75)	54	35	8	2
Cholesterol, g/kJ	4.7	<0.0001	- ^f^	- ^f^	0.63 (0.55–0.70)	47	39	13	2
Carbohydrate, g/kJ	−1.8	0.0030	−15	18	0.69 (0.62–0.74)	51	37	10	2
Total sugar, g/kJ	0.6	0.2410	−28	54	0.68 (0.62–0.74)	51	38	10	1
Fibre, g/kJ	−3.1	0.0020	−34	43	0.74 (0.69–0.78)	53	39	7	1
Alcohol, g/kJ ^b,g^	0	<0.0001	- ^f^	- ^f^	0.80 (0.74–0.84)	56	37	6	1
Sodium, mg/kJ	−1.6	0.1560	−27	34	0.60 (0.53–0.67)	47	40	10	3
Potassium, mg/kJ	−0.3	0.3990	−22	28	0.69 (0.63–0.74)	50	39	9	1
Calcium, mg/kJ	−2.5	0.0290	−41	62	0.72 (0.66–0.77)	56	35	7	1
Magnesium, mg/kJ	−1.3	<0.0001	−19	21	0.78 (0.73–0.82)	53	40	6	0
Phosphorus, mg/kJ	−0.7	0.0930	−24	33	0.69 (0.63–0.75)	53	35	10	2
Iron, mg/kJ	−0.9	0.0740	−20	24	0.72 (0.65–0.76)	53	37	8	1
Copper, mg/kJ	−5.7	0.0010	−53	96	0.70 (0.64–0.75)	51	39	7	2
Zink, mg/kJ	0.5	0.3790	−21	27	0.66 (0.59–0.71)	49	37	12	1
Selenium, µg/kJ	1.9	0.0060	- ^f^	- ^f^	0.64 (0.57–0.70)	51	36	10	2
Iodine, µg/kJ	−3.4	0.0230	- ^f^	- ^f^	0.64 (0.57–0.70)	46	41	11	2
Retinol, µg/kJ	3.7	0.0070	- ^f^	- ^f^	0.67 (0.61–0.73)	50	39	10	2
Beta carotene, µg/kJ	−1.4	0.4840	−65	157	0.65 (0.58–0.72)	50	38	11	1
Vitamin A, µg/kJ	4.8	0.1030	−50	102	0.52 (0.44–0.60)	43	39	15	4
Vitamin D, µg/kJ	4.5	0.0010	- ^f^	- ^f^	0.65 (0.59–0.71)	51	37	11	1
Vitamin E, mg/kJ	0.1	0.5060	−34	50	0.75 (0.69–0.79)	53	37	8	1
Vitamin K, µg/kJ ^b^	−0.06	0.0290	- ^f^	- ^f^	0.69 (0.63–0.74)	51	36	11	1
Thiamine, mg/kJ	−1.4	0.0310	−23	32	0.63 (0.56–0.69)	48	37	13	2
Riboflavin, mg/kJ	0.3	0.9000	−33	45	0.73 (0.68–0.78)	54	38	7	1
Niacin, mg/kJ	2.2	0.0260	−26	46	0.70 (0.64–0.76)	51	38	10	2
Vitamin B6, mg/kJ	−0.3	0.6560	- ^f^	- ^f^	0.63 (0.55–0.69)	50	37	10	3
Vitamin B12, µg/kJ	3.1	0.0160	- ^f^	- ^f^	0.61 (0.54–0.67)	49	35	13	3
Vitamin C, mg/kJ	−1.3	0.2290	−55	87	0.64 (0.57–0.70)	49	38	10	3
Folate, µg/kJ	−0.2	0.3690	−39	48	0.68 (0.61–0.74)	53	36	9	2

^a^ Based on log-transformed density intakes. ^b^ Based on raw intake density intakes. ^c^ Median bias is reported as a percentage for log-transformed density intakes and unit difference for raw density intakes. ^d^ *p*-value, the test of difference in intake between (log^a^) FFQ_baseline_ and (log^a^) mean of three 24-HDRs using the Wilcoxon signed-rank test. ^e^ Bland–Altman limits of agreement (LOAs) are reported as a percentage difference. ^f^ Bland–Altman limits of agreement (LOAs) are not reported as LOAs depending on the level of the nutrient. ^g^ Alcohol had a large proportion of non-consumers.

**Table 7 nutrients-15-02389-t007:** Food group intakes from the FFQ_baseline_ compared with the FFQ_12 months_ for median bias, Bland–Altman LOAs, Spearman’s correlation coefficient, and cross-classification (*n* = 415).

Food Groups ^a^	Median Bias ^b^	*p*-Value ^d^	Bland–Altman LOA ^e^		Spearman’s Correlation Coefficient	Cross-Classification			
			Lower	Upper	(95% CI)	Same Quartile (%)	Adjacent Quartile (%)	Opposite Quartile (%)	Extreme Opposite Quartile(%)
Fruits, g	−12.26	<0.0001	−66	123	0.74 (0.68–0.79)	51	41	6	1
Vegetables, g	−8.79	<0.0001	−54	95	0.73 (0.67–0.77)	52	39	9	1
Potatoes, g	−11.11	0.0003	−68	132	0.69 (0.62–0.74)	51	39	9	1
Legumes, g	- ^c^	0.8210	- ^c^	- ^c^	0.75 (0.70–0.80)	53	36	11	1
Whole grains, g ^f^	−9.93	0.0004	−74	228	0.62 (0.55–0.69)	46	40	11	3
Eggs, g	−10.44	<0.0001	−72	147	0.61 (0.53–0.67)	48	36	13	2
Poultry, g	- ^c^	<0.0001	- ^c^	- ^c^	0.60 (0.52–0.66)	46	36	16	2
Red meat, g	- ^c^	<0.0001	- ^c^	- ^c^	0.76 (0.70–0.81)	56	36	7	1
Processed red meat, g	- ^c^	<0.0001	- ^c^	- ^c^	0.78 (0.73–0.82)	57	35	7	0
Fast food, g	- ^c^	0.1363	- ^c^	- ^c^	0.75 (0.69–0.80)	55	37	7	1
Fish and seafood, g	- ^c^	<0.0001	- ^c^	- ^c^	0.75 (0.70–0.80)	53	38	7	1
Dairy products, g	−7.4	0.0005	−83	281	0.76 (0.70–0.80)	57	35	7	1
Fermented dairy products, g	−12.36	<0.0001	−78	204	0.72 (0.66–0.77)	54	37	8	1
Fat, g	−8.05	<0.0001	−67	81	0.64 (0.57–0.70)	45	42	11	2
Soft drinks, g	- ^c^	0.0907	- ^c^	- ^c^	0.80 (0.75–0.84)	63	31	5	2
Coffee, g	- ^c^	0.0025	- ^c^	- ^c^	0.78 (0.73–0.83)	60	33	7	0
Tea, g	- ^c^	<0.0001	- ^c^	- ^c^	0.88 (0.84–0.90)	69	27	3	0

^a^ Based on log-transformed intakes. ^b^ Median bias is reported as a percentage. ^c^ Percentage median bias and LOAs are not reported due to zero-intake. ^d^ *p*-value, the test of difference in intake between (log^a^) FFQ_baseline_ and (log^a^) FFQ_12 months_ using the Wilcoxon signed-rank test. ^e^ Bland–Altman limits of agreement (LOAs) are reported as a percentage difference. ^f^ Estimated whole-grain intake from whole-grain products.

## Data Availability

Data may be available upon request to the Danish Cancer Society (contact: dchdata@cancer.dk) and if approved by the DCH-NG project committee.

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
