# Peer review of "Relative Validity and Reproducibility of a Web-Based Semi-Quantitative Food Frequency Questionnaire in the Danish Diet, Cancer, and Health—Next Generations MAX Study"

_nutrients, 2023, doi:10.3390/nu15102389_

Round 1

Reviewer 1 Report

This paper assesses the new  web-based Food Frequency Questionnaire (FFQ) for both validity and reproducibility.  Spearman rank correlations between the mean of three 24hr dietary recalls (HDRs) and the FFQ are used for validity, and FFQ at baseline and 12 months for reproducibility.  Cross-validation into quartiles is also reported.  In general the paper is well written, and thought has been given to the appropriateness of analyses needed.  I have some comments below which mainly refer to clarity of methods.

Comments

Regarding reference [1].  This is collection of articles of varying topics and with varying authorships.  Even it were the same author contribution throughout, it would still be useful to be more specific in regards to what is being referenced.  I assume that most references are to "Willett, W., Elizabeth L., 'Reproducibility and Validity of Food Frequency Questionnaires', Nutritional Epidemiology, 3rd Ed, Monographs in Epidemiology and Biostatistics (2012)" but there are likely references to other parts of the collection too.  Please provide these as separate references in the references and with the complete author list.

I agree that in [1], it is suggested that three HDRs are usually appropriate, but is using the mean of the HDRs the best was to carry out the analysis?  It seems that while this is sometimes done, it is not always the case and so is limiting the analysis to just the mean HDR suitable here?  Maybe it is, but justifying this would be useful.   

Spearman's rank correlation was used, which when used as a measure of validity and reproducibility, can be large even when the relationship between variables is very non-linear.  That said, I agree that Spearman correlation is likely appropriate.  However I think it is also best to justify the use of Spearman rank correlation over Pearson correlation following log transformations (e.g., [1] uses Pearson but does note that Spearman can be used when variables are non-normal).  Also, many seem to use Pearson correlations, and so it would be good to know what differences there are, which would then be useful to compare with findings from other studies (many of which would have similarly distributed variables). 

Cross-validation - it would be to explain how this was done.  There are different ways to conduct cross-validation.

Figure 3 seems important, and yet it is hardly discussed.  Also, is there a missing reference to this figure on line 265 (page 8), where is says "Figure and Supplementary Figure S1"?  I think it would be good to discuss what is depicted in the figure more, and perhaps also highlight that the "fanning" is due to back-transformation to the original scale from the log-scale.

Do the linear trends in Figure S1 warrant further attention/discussion?  Linear trends in Bland-Altman plots usually do.

Some proof reading required, e.g., missing reference, "diving" instead of "dividing" page 6.

Reviewer 2 Report

Congratulations to the authors.Interesting research to address the measurement of food intake that allows knowing the nutrients density and the foods that compose it. both describe methods have a strength and weaknesses as shown by thr results obtained.Howevwe it is important to have the range of variability in the difference obtained over the average to apply and analyze results. The study of diet is complex in population studies . These results ar a great contribution to continue discussing the methods to know the composition of the diet and how to record it Perhaps one of the challenges is to define the critiacl foods to analyze and evaluate in both metods. Perhaps aad in results that these methods show trends both in te composition of the food and in the nutritional sufficciency of macro and micronutrients
